# m* of two-dimensional electron gas: A neural canonical transformation study

Hao Xie[1,2], Linfeng Zhang[3,4*] and Lei Wang[1,5†]

**1** Institute of Physics, Chinese Academy of Sciences, Beijing 100190, China
**2** University of Chinese Academy of Sciences, Beijing 100049, China
**3** DP Technology, Beijing, 100080, China
**4** AI for Science Institute, Beijing, 100080, China
**5** Songshan Lake Materials Laboratory, Dongguan, Guangdong 523808, China

★ linfeng.zhang.zlf@gmail.com , † wanglei@iphy.ac.cn

## Abstract

The quasiparticle effective mass $m^*$ of interacting electrons is a fundamental quantity in the Fermi liquid theory. However, the precise value of the effective mass of uniform electron gas is still elusive after decades of research. The newly developed neural canonical transformation approach [Xie *et al.*, J. Mach. Learn. 1, (2022)] offers a principled way to extract the effective mass of electron gas by directly calculating the thermal entropy at low temperature. The approach models a variational many-electron density matrix using two generative neural networks: an autoregressive model for momentum occupation and a normalizing flow for electron coordinates. Our calculation reveals a suppression of effective mass in the two-dimensional spin-polarized electron gas, which is more pronounced than previous reports in the low-density strong-coupling region. This prediction calls for verification in two-dimensional electron gas experiments.

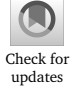

# 1   Introduction

Landau's Fermi liquid theory [1] is one of the cornerstones of condensed matter physics [2]. It explains the mystery why the non-interacting picture can largely apply to real metals despite the strong Coulomb repulsion between electrons. The essence is that a Fermi liquid consists of quasiparticles that are adiabatically connected to bare electrons. Such a renormalization procedure can be encapsulated in only a handful of parameters, from which one can predict a broad range of physical properties of the system. One such parameter is the quasiparticle effective mass $m^*$, which is the central focus of this work.

Surprisingly, the precise value of the quasiparticle effective mass of uniform electron gas is still controversial after more than fifty years of research [3–17]. The uniform electron gas consists of electrons distributed homogeneously in a background of positive charges. Despite of being simple, the model captures the essence of electron correlation effects and serves as a foundational model of interacting electrons [18, 19].

Depending on the spatial dimension and spin polarization of the uniform electron gas, previous results may differ quantitatively or even qualitatively on whether the quasiparticle effective mass is enhanced ($m^*/m > 1$) or suppressed ($m^*/m < 1$) compared to the bare electron mass $m$. Resolving these discrepancies within the same approach can be challenging. For example, there lacks a systematic way of improving various approximate analytic calculations to reach a consensus [3–9]. It is hoped that numerical calculations offer more reliable predictions to the effective mass. However, two recent quantum Monte Carlo (QMC) studies [16, 17] report drastically different effective masses for the three-dimensional electron gas. The reason for such discrepancy is unclear and may be related to different (but equivalent) ways of defining the effective mass as well as different approximations employed in the methods. The situation is also not clear in the two-dimensional case, even if one employs the same kind of QMC method [12–15]. There, the predicted effective mass differ qualitatively depending on how to process the QMC data.[1] These discrepancies are related to ambiguities in relating excited state energies to the effective mass [19], which again entangle with approximations and finite size errors in the calculations.

Resolving the discrepancy on the effective mass of uniform electron gas is not only a theoretical question with pure academic interests, but also of direct experimental relevance. One can measure the effective mass in a semiconductor quantum well via quantum oscillations [20–24] or thermodynamics [25], which is a high-quality realization of the two-dimensional electron gas (2DEG) with tunable densities. Unless otherwise specified, we will focus on the spin-polarized case in this paper, which can be conveniently realized in experiments by applying an in-plane magnetic field.

---

[1] In fact, the predictions from each group also evolve over years; see Refs. [10–12] and [13–15] respectively.

At sufficiently low temperature, the entropy per particle of 2DEG $s/k_B = \frac{\pi^2}{3}\frac{m^*}{m}\frac{T}{T_F}$ exhibits linear dependence on the temperature $T$, where $k_B$ is the Boltzmann constant and $T_F$ is the Fermi temperature. Therefore, one can directly estimate the effective mass from the entropy ratio of interacting ($s$) and non-interacting ($s_0$) electron gases [26]:

$$\frac{m^*}{m} = \frac{s}{s_0}. \tag{1}$$

By direct access of the thermodynamic observables, one can avoid subtleties in relating excitation energies of finite size system to the quasiparticle effective mass [19]. However, previous finite-temperature calculations of the uniform electron gas do not resolve the issue related to the effective mass [26] because they focus on melting of the Wigner crystal [27] or the equation of state in the warm dense matter region [28–30], both are outside the scope of Fermi-liquid-like behavior. This is partially due to the fact that the adopted QMC methods typically suffer less from the sign problem at low density and high temperature, where the fermionic nature of the system is less pronounced.

In this paper, we employ the recently developed neural canonical transformation approach [31] to study 2DEG at low temperature and estimate the effective mass via the entropy ratio Eq. (1). Neural canonical transformation leverages recent advances in deep generative models [32] for variational free energy calculation of interacting fermions at finite temperature. This approach is particularly suitable for the present task for two reasons: firstly, the employed variational density matrix ansatz fits nicely to the philosophy of Fermi liquid theory; secondly, the thermal entropy can be directly accessed unlike other conventional QMC methods.

Consider $N$ electrons in a two-dimensional periodic box of length $L$. We set the energy unit to be Rydberg $\hbar^2/2ma_0^2$, where $a_0 = \hbar^2/me^2$ is the Bohr radius. The dimensionless Wigner-Seitz parameter $r_s = L/(\sqrt{\pi N}a_0)$ measures the average distance between electrons in the unit of Bohr radius. The Hamiltonian reads [33]

$$H = -\frac{1}{r_s^2}\sum_{i=1}^{N}\nabla_i^2 + \frac{2}{r_s}\sum_{i<j}^{N}\frac{1}{|\boldsymbol{r}_i - \boldsymbol{r}_j|} + \text{const.}, \tag{2}$$

where $\boldsymbol{r}_i = (x_i, y_i)$ is the coordinate of the $i$-th electron. The constant term in Eq. (2) refers to the energy due to the neutralizing background.

## 2 Method

To investigate the finite-temperature properties of Eq. (2), we minimize the variational free energy

$$F = \frac{1}{\beta}\text{Tr}(\rho\ln\rho) + \text{Tr}(\rho H) \tag{3}$$

with respect to a many-electron density matrix $\rho$, where $\beta = 1/k_B T$ is the inverse temperature. In practice, $T$ is measured relative to the Fermi temperature $k_B T_F = 4\text{Ry}/r_s^2$. The variational free energy Eq. (3) is lower bounded by the true free energy of the system, i.e., $F \geq -\frac{1}{\beta}\ln Z$, where $Z = \text{Tr}e^{-\beta H}$ is the partition function. The equality holds only when the variational density matrix coincides with the exact one, i.e., $\rho = e^{-\beta H}/Z$.

The variational density matrix is expressed as a weighted sum over a family of many-body orthonormal basis states

$$\rho = \sum_K p(K)|\Psi_K\rangle\langle\Psi_K|. \tag{4}$$

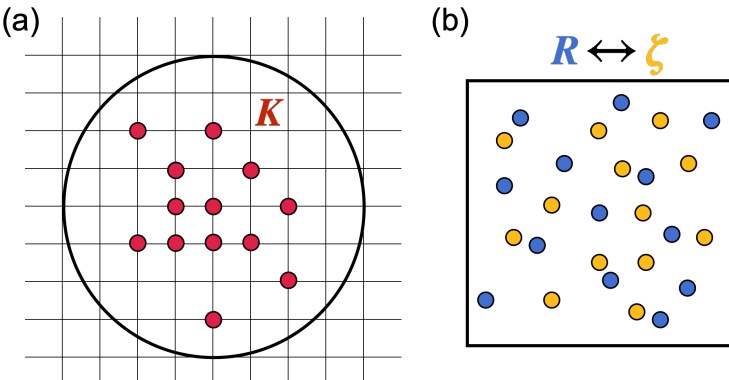

Figure 1: (a) A set of occupied momenta $K$ residing on the reciprocal space grid within an energy cutoff indicated by the circle. In the figure there are $N = 13$ electrons distributed in $M = 49$ allowed momenta. We use the autoregressive network Eq. (6) to model the Boltzmann distribution of such configurations. (b) A set of electron coordinates $R$ residing in a periodic box. A normalizing flow network transforms $R$ to a new set of quasiparticle coordinates $\zeta$ in the same box. This induces a unitary transformation which, when acting upon the plane-wave Slater determinants, produces a set of many-body basis states Eq. (8).

Here $K \equiv \{k_1, k_2, \ldots, k_N\}$ represents a *set* of occupied momenta, each (under the periodic boundary conditions) taking one of the discrete values $k = \frac{2\pi n}{L}$ ($n \in \mathbb{Z}^2$) without duplication as required by the Pauli exclusion principle. Such setting is closely in line with an essential point of Fermi liquid theory [1]: one can label the low-energy excited states using the same quantum number as the ideal Fermi gas. In practice, one has to truncate the momenta within an energy cutoff, which is set to be sufficiently large to avoid bias in the considered temperature range. Therefore, for $M$ possible momenta within the energy cutoff, the summation Eq. (4) involves $\binom{M}{N}$ terms. See Fig. 1(a) for a schematic illustration.

Substituting the density matrix ansatz Eq. (4) into Eq. (3), one finds an unbiased estimator for the variational free energy:

$$F = \underset{K \sim p(K)}{\mathbb{E}} \left[ \frac{1}{\beta} \ln p(K) + \underset{R \sim |\Psi_K(R)|^2}{\mathbb{E}} \left[ E_K^{\text{loc}}(R) \right] \right]. \tag{5}$$

Here $R \equiv \{r_1, r_2, \ldots, r_N\}$ is the set of electron coordinates and $\Psi_K(R) = \langle R | \Psi_K \rangle$ the corresponding basis wavefunction. The local energy is defined as $E_K^{\text{loc}}(R) = -\frac{1}{r_s^2} \sum_i \left[ \nabla_i^2 \ln \Psi_K(R) + (\nabla_i \ln \Psi_K(R))^2 \right] + \frac{2}{r_s} \sum_{i<j} \frac{1}{|r_i - r_j|} + \text{const}$. We use Ewald summation to evaluate the Coulomb interaction term, while the gradient and Laplacian operator appearing in the kinetic term can be computed using automatic differentiation. We model the Boltzmann distribution $p(K)$ and wavefunction $\Psi_K(R)$ using two generative networks.

We use a variational autoregressive network [34, 35] to model the normalized Boltzmann distribution $p(K)$ over a set of discrete momenta:

$$p(K) = \prod_{i=1}^{N} p(k_i | k_{<i}), \tag{6}$$

where each factor in the product is a parametrized conditional probability. To facilitate the sampling of these conditional probabilities, we assign a unique index $\texttt{idx}(k) \in \{1, 2, \ldots, M\}$ to each of the $M$ available momenta, e.g., according to their single-particle energies.[2] We

---

[2]In practice, we choose to arrange the momenta in the order of *decreasing* energy. This stems from the observa-

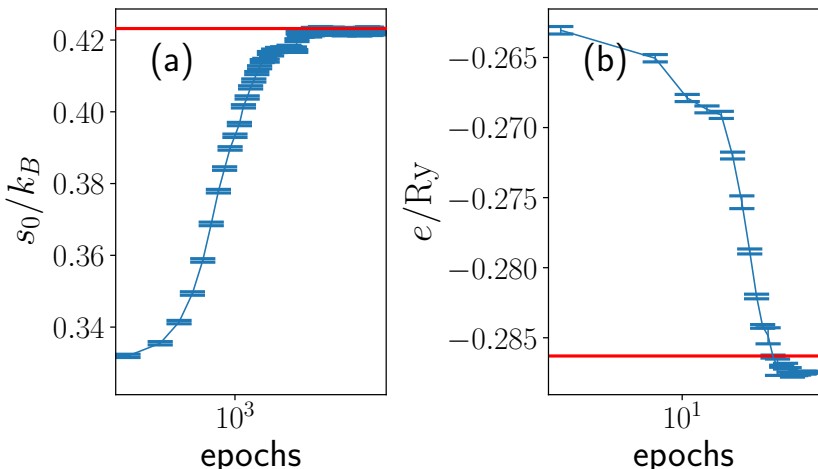

Figure 2: Two limiting cases of the present approach. The red horizontal lines indicate benchmark values. (a) Non-interacting limit. The entropy per particle of $N = 37$ free electrons at $T/T_F = 0.15$. The converged result $0.4227(4)$ agrees well with the exact value $0.4232$ calculated by recursion in the canonical ensemble [39].[3] (b) Zero-temperature limit. The ground-state energy per particle of $N = 37$ electrons at $r_s = 5$. The converged result $-0.28746(8)$ is lower than the variational Monte Carlo result $-0.2863(1)$ of Ref. [40].

then model $p(K)$ using a neural network that maps the set $K = \{k_1, k_2, \ldots, k_N\}$ to $N$ vectors $\hat{k}_1, \hat{k}_2, \ldots, \hat{k}_N$, where $\hat{k}_i \in \mathbb{R}^M$ denotes the conditional log-probability of $\texttt{idx}(k_i)$ given $k_{<i}$. To ensure the autoregressive property, i.e., $\hat{k}_i$ depends only on $k_j$ with $j < i$, we implement the network as a transformer with causal self-attention layers [36]. In addition, to accommodate the Pauli principle, we require Eq. (6) assign nonzero probabilities only to those momentum configurations satisfying $\texttt{idx}(k_1) < \texttt{idx}(k_2) < \ldots < \texttt{idx}(k_N)$. This can be achieved by carefully masking out disallowed configurations in the output logits $\hat{k}_i$.[3] We note that Refs. [37,38] devised an alternative autoregressive model in the bit string representation with fixed number of nonzero elements.

Using the autoregressive model Eq. (6) rather than enumerating all possible excitations [31] in the summation Eq. (4) allows us to incorporate a combinatorially large number of many-body states and access broader temperature range. One can estimate the thermal entropy unbiasedly via the estimator

$$s = -\frac{1}{N} \mathop{\mathbb{E}}_{K \sim p(K)} [\ln p(K)]. \tag{7}$$

Note such a simple and tractable expression for the entropy is a direct consequence of orthonormality of the many-body basis $|\Psi_K\rangle$.

Next, to parametrize a family of orthonormal many-body states $|\Psi_K\rangle$, we perform a unitary transformation on the basis of plane-wave Slater determinants. In practice, we construct the

---

tion that a random parameter initialization of the present autoregressive network tends to assign higher probability to configurations with large indices, which then have large overlap with the target Boltzmann distribution.

[3]See the Appendix for (a) additional benchmark for three-dimensional spin-polarized uniform electron gas; (b) the analytic method to compute the thermal entropy of non-interacting Fermi gas in the canonical ensemble; (c) discussion and implementation details about the use of twist-averaged boundary conditions; details on (d) model architectures and (e) the training procedure.

unitary transformation as a learnable bijection from the electron coordinates $\boldsymbol{R}$ to a new set of *quasiparticle coordinates* $\boldsymbol{\zeta}$, as illustrated in Fig. 1(b). The wavefunction reads [31]

$$\Psi_{\boldsymbol{K}}(\boldsymbol{R}) = \frac{\det\left(e^{i\boldsymbol{k}_i\cdot\boldsymbol{\zeta}_j/L}\right)}{\sqrt{N!}} \cdot \left|\det\left(\frac{\partial\boldsymbol{\zeta}}{\partial\boldsymbol{R}}\right)\right|^{\frac{1}{2}}. \tag{8}$$

The originally non-interacting plane waves possessed by individual electrons would interfere with each other due to correlation effects introduced by the coordinate transformation. Such a picture closely mimics the renormalization process as depicted in Fermi liquid theory. Technically, Eq. (8) differs from the standard Slater-Backflow trial wavefunction by the presence of an additional Jacobian determinant. This factor turns out to play a crucial role of preserving orthonormality of the basis $\int d\boldsymbol{R}\, \Psi_{\boldsymbol{K}}^*(\boldsymbol{R})\Psi_{\boldsymbol{K}'}(\boldsymbol{R}) = \delta_{\boldsymbol{K}\boldsymbol{K}'}$. Note also that the state Eq. (8) involves coordinate transformation in a many-body context [41], where one needs to deal with the extra issue of permutation equivariance compared to the single-particle setting [42–44].

We use a normalizing flow [45] to implement the bijective map between the electron and quasiparticle coordinates. This can be regarded as a generalization of the backflow transformation with invertible neural networks [46]. We compose the FermiNet [47] blocks into a residual network to carry out permutation and translation equivariant transformation of electron coordinates. We also modify the electron distance features to comply with the periodic nature of the simulation box.[3]

It is instructive to examine the present approach in two limiting cases. Firstly, in the non-interacting limit the problem reduces to a classical statistical mechanics problem: one needs to distribute $N$ particles in $M$ possible momenta according to the probability distribution $p(\boldsymbol{K})$ to minimize the free energy $F = \mathbb{E}_{\boldsymbol{K}\sim p(\boldsymbol{K})}\left[\frac{1}{\beta}\ln p(\boldsymbol{K}) + \sum_{i=1}^{N}\frac{\hbar^2\boldsymbol{k}_i^2}{2m}\right]$, where the second term is the non-interacting energy. In this case, one can trivially set the normalizing flow to an identity map and optimize only the autoregressive network. Figure 2(a) shows a typical training process, where the entropy Eq. (7) steadily approaches the exact value [39]. Note the calculation of exact non-interacting entropy in the canonical ensemble is not a completely trivial task.[3] Secondly, in the zero-temperature limit, $p(\boldsymbol{K})$ is nonzero only for one particular momentum configuration $\boldsymbol{K}_0$ corresponding to the closed-shell non-interacting ground state. The present approach then reduces to the usual ground-state variational Monte Carlo method. As an example, Fig. 2(b) shows the optimized ground-state energy density $e = \frac{1}{N}\mathbb{E}_{\boldsymbol{R}\sim\left|\Psi_{\boldsymbol{K}_0}(\boldsymbol{R})\right|^2}\left[E_{\boldsymbol{K}_0}^{\text{loc}}(\boldsymbol{R})\right]$ for a particular set of system parameters, which is lower than previous report using the Slater-Jastrow ansatz [40]. In general cases, one has to jointly optimize the autoregressive model and the normalizing flow. We show some additional benchmark results for the three-dimensional spin-polarized electron gas in the Appendix.[3]

To understand how variational free energy calculation reveals the quasiparticle effective mass, note the effective mass affects low-temperature thermodynamics of the system via the density of states of low-lying excitations. In practice, we pretrain the state occupation $p(\boldsymbol{K})$ using non-interacting energies as in Fig. 2(a). Thus, $p(\boldsymbol{K})$ will initially give the same entropy of the ideal Fermi gas. We initialize the normalizing flow network to be close to an identity map. Training of the normalizing flow will modify the many-body basis Eq. (8) and thus change the quasiparticle energy spacing and density of states. On the other hand, the autoregressive model will also adjust the Boltzmann distribution accordingly, causing the entropy to depart from the initial non-interacting value. Putting it all together, the entropy ratio Eq. (1) thus provides a principled way to extract the effective mass from the quasiparticle energy spectrum.

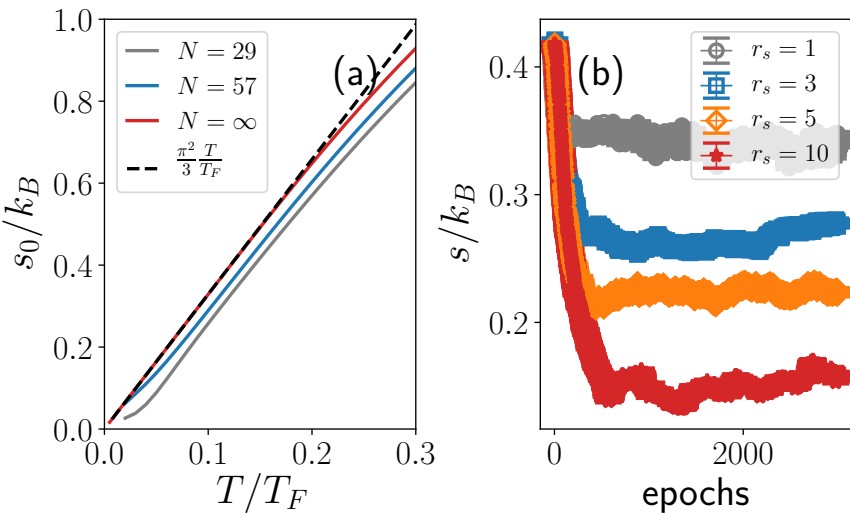

Figure 3: (a) Entropy per particle of two-dimensional ideal Fermi gas for several electron numbers $N$, including the thermodynamic limit. The dashed line shows the linear behavior in the low-temperature limit. For finite $N$, we have used the twist-averaged boundary conditions to reduce the finite size effect. (b) Entropy per particle as a function of training epochs for $N = 29$ electrons at $T/T_F = 0.15$ for various densities $r_s$. The reduction of entropy from the initial non-interacting value indicates suppression of the quasiparticle effective mass.

## 3 Results

To access the quasiparticle effective mass of electron gas via the entropy ratio Eq. (1), one should consider temperatures $T$ well below the Fermi temperature $T_F$. Figure 3(a) shows the entropy per particle of ideal Fermi gas in the thermodynamic limit $N = \infty$,[4] which exhibits linear behavior $s_0/k_B = \frac{\pi^2}{3} \frac{T}{T_F}$ in the low-temperature limit and crosses over to $s_0/k_B = 2 + \ln \frac{T}{T_F}$ in the high-temperature limit. On the other hand, the temperature should also not be too low in practical calculations, otherwise the finite size effect would cause the entropy to deviate from the ideal linear behavior due to a small energy scale $\hbar^2/mL^2 \sim N^{-1}$, as shown in Fig. 3(a) for $N = 29$ and 57 non-interacting electrons. We choose to set $T/T_F = 0.15$ to balance these two considerations.

To obtain conclusive predictions of the effective mass, we adopt the twist-averaged boundary conditions [48] to alleviate the finite size effect.[3] Moreover, one can reasonably expect that by taking the entropy ratio Eq. (1), the remaining finite size errors involved in the interacting and non-interacting systems would further cancel out. Fig. 3(b) shows the interacting entropy as a function of training epochs for $N = 29$ and various densities $r_s$. One clearly sees the entropies are reduced from the initial non-interacting values, indicating a suppression of the effective mass upon increasing $r_s$. The entropy fluctuates more strongly than the free energy since it is more sensitive to the variation of model parameters in the training.

Early analytical calculations [7,49] find enhanced or non-monotonic $r_s$-dependence of the effective mass in spin-polarized 2DEG. On the other hand, several QMC calculations [13–15] consistently find monotonically suppressed effective mass as $r_s$ increases, but the quantitative

---

[4]For two-dimensional ideal Fermi gas in the thermodynamic limit, one has $s/k_B = 2f_2(z)/f_1(z) - \ln z$ and $T/T_F = 1/f_1(z)$. Here $0 \le z < \infty$ is the fugacity and $f_\nu(z) = z - z^2/2^\nu + z^3/3^\nu - \cdots$ is the Fermi-Dirac function.

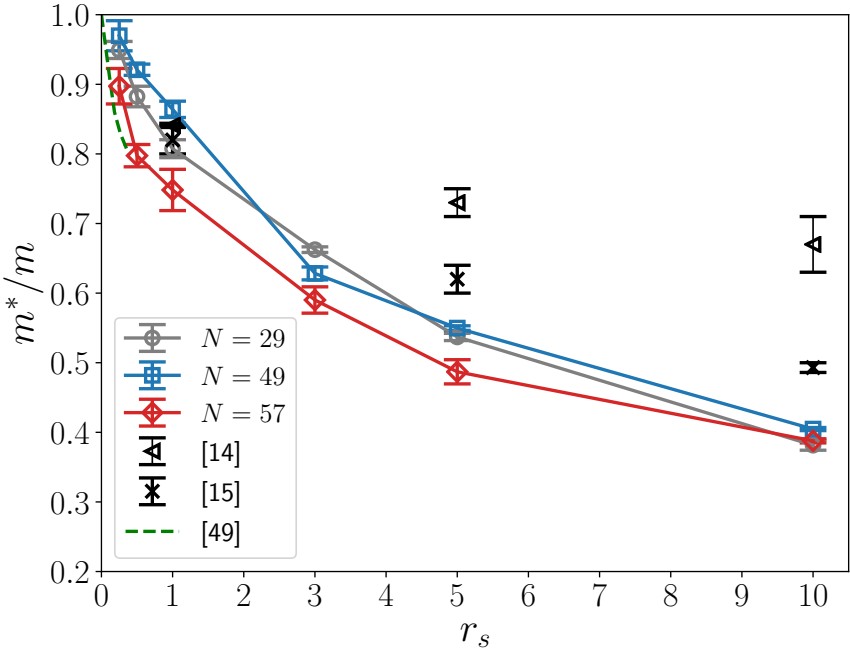

Figure 4: Quasiparticle effective mass computed using the entropy ratio Eq. (1) for $N = 29, 49$ and 57 electrons at $T/T_F = 0.15$. Also shown for comparison are the QMC data [14, 15] extrapolated to the thermodynamic limit, as well as the analytic result [49] which is valid in the weak-coupling limit $r_s \to 0$.

predictions still differ, especially for large $r_s$ as shown in Figure 4.[5] The discrepancy is due to different ways of extracting effective mass from the excitation energies. Refs. [13, 14] obtain the effective mass by differentiating the fitted energy band, while Ref. [15] is based on its relation to other Landau Fermi liquid parameters by Galilean invariance. Both approaches possess a number of uncertainties, such as the fitting range of momentum space and integration error in estimating the Fermi liquid parameters. Though, the authors of Ref. [15] appeared to be more confident with the larger effective mass values reported in Refs. [13, 14].

Figure 4 also shows our estimates of the effective mass for $N = 29, 49$ and 57 electrons based on the entropy ratio Eq. (1). We extract the interacting entropy by performing an exponentially-weighted moving average over the training epochs. The error bars take into account both the statistical uncertainties due to Monte Carlo sampling and the fluctuation of variational parameters due to noisy gradients.[6] The computed effective mass decreases monotonically with increasing interaction strength. In the small $r_s$ region, the predicted effective mass appears to converge well to the analytical result reported in Ref. [49], shown as the green dashed line in Fig. 4, which is reliable in the weak-coupling limit $r_s \to 0$. However, our predictions are lower than previous QMC results [14, 15] when $r_s$ is large. Such differences cannot be attributed to the remaining finite size errors.[3] Since the present approach based on the entropy ratio Eq. (1) is less subject to *ad hoc* assumptions and data processing

---

[5] Data taken from table IV of Ref. [15]. These results are extrapolated to the thermodynamic limit based on the data of $N = 29, 57$ and 101 electrons with a presumed $N$-scaling dependence. The data of Ref. [14] are improved upon that of Ref. [13] so we only plot the former.

[6] See https://github.com/fermiflow/CoulombGas for code written in Jax [81]. The repository also contains the original data, trained models and data processing scripts that can be used to reproduce the main results in Fig. 4.

schemes, we believe it offers a cleaner and more reliable prediction on the effective mass of spin-polarized 2DEG.

On the experimental side, both enhanced [50] and later then suppressed [51,52] effective mass of the spin-polarized 2DEG are reported in different systems. The discrepancy was attributed to the valley degeneracy [53,54] involved in the sample used in Ref. [50]. The experimental data [51,52] spread widely between our predictions and those of Refs. [13–15].[7] Confirmation of the present results calls for a new generation of experimental efforts, where besides data uncertainty issue one also has to account for various complications in reality for a fair comparison, such as thickness of the electron layer, disorder and temperature effects [7, 55–57].

## 4  Discussions

The variational approximation of the present approach may be improved by adopting alternative network ansatz [58–60] and optimization schemes [61, 62]. We have documented the original data and trained models in the code repository[6] to facilitate further developments. In the present implementation, the finite size errors have been largely reduced by adopting the twist-averaged boundary conditions. To scale up the calculation to larger systems one can employ machine learning techniques such as gradient checkpointing [63] and distributed training [64]. Specifically, techniques for efficiently training flow models and invertible neural networks [65–70] can be useful. It is also profitable to integrate the present standalone implementation into an existing software framework [71]. On the other hand, a rigorous finite-size scaling theory for the entropy of uniform electron gas is also valuable for a direct extrapolation to the thermodynamic limit.

With suitable extensions of the model architecture, the technique developed in this paper can apply equally well to the spin-unpolarized case. This may shed new light on the conflicting results reported in the literature on the three-dimensional [16,17] and two-dimensional [10–15] electron gases. While we have been focusing on the quasiparticle effective mass, the outcomes of the present approach are also directly relevant to the exchange-correlation free energy, which are useful for the thermal density functional theory [26,30] and thermodynamic measurements in the 2DEG [25]. Along this line, it is also possible to extend the present study to the grand canonical ensemble and compute the compressibility and susceptibility of electron gas measurable in experiments [22,23,72,73]. Finally, having direct access to the energy and wavefunction of low-lying excitations may also allow for calculation of spectral functions at real frequencies.

Neural canonical transformation [31] not only serves as a variational free energy approach powered by deep learning techniques, but also nicely incorporates basic notions of Fermi liquid theory. For example, the probabilistic model $p(K)$ in Eq. (4) actually encapsulates the Landau's energy functional for quasiparticles. Moreover, the unitary transformation implemented as a normalizing flow between the electron and quasiparticle coordinates vividly illustrates the notion of adiabatic continuity when switching on interactions [2]. Because of these technical and physical considerations, we are optimistic with the outcome of applying neural canonical transformation to a broader class of interacting fermion problems.

---

[7]We note that the indicated $r_s$ values of the same data plotted in Fig. 2 of Ref. [49] and Fig. 9 of Ref. [14] differ by a factor of two.

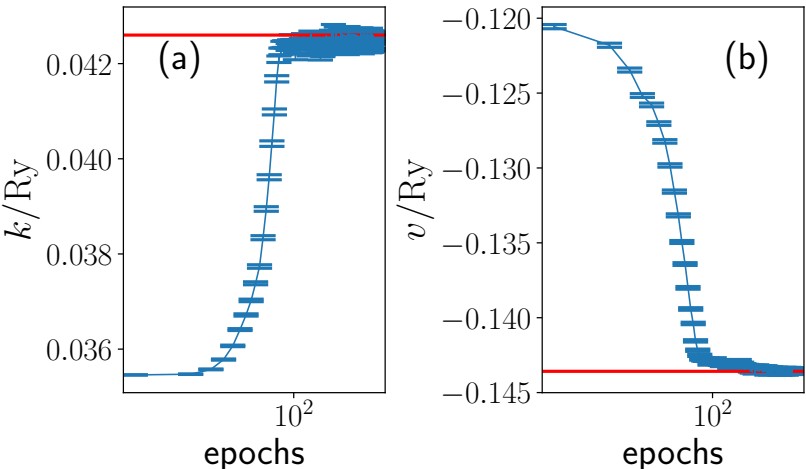

Figure S1: (a) Kinetic energy and (b) potential energy per particle of $N = 33$ spin-polarized electrons in 3D with $r_s = 10$ and $T/T_F = 0.0625$. The converged values are $k = 0.04250(7)$ and $v = -0.14360(7)$, while the red horizontal lines $k = 0.0426(1)$ and $v = -0.14358(1)$ are from the restricted path integral Monte Carlo calculation [28].

## Acknowledgements

We thank Yuan Wan, Pan Zhang, Kun Chen, Xinguo Ren, and Giuseppe Carleo for useful discussions. This project is supported by the Strategic Priority Research Program of the Chinese Academy of Sciences under Grant No. XDB30000000 and National Natural Science Foundation of China under Grant Nos. 92270107, 12188101, T2225018, and T2121001.

## A  Benchmark for three-dimensional spin-polarized uniform electron gas

We carry out additional benchmark calculation for three-dimensional spin-polarized uniform electron gas in a periodic cubic box of length $L$. We use the same network architectures and training procedure as described in the main text, except that $r_s = \left(\frac{3}{4\pi N}\right)^{\frac{1}{3}} \frac{L}{a_0}$ and $k_B T_F = \left(\frac{9\pi}{2}\right)^{\frac{2}{3}} \mathrm{Ry}/r_s^2$ in the three-dimensional case. The two panels of Fig. S1 display a typical training process of the kinetic energy $k$ and potential energy $v$ per particle, respectively. Note that $r_s = 10$ is within the low-density parameter range where the restricted path integral Monte Carlo method [28] can favorably produce accurate results.

## B  Entropy of non-interacting fermions in the canonical ensemble

To compute the entropy of $N$ non-interacting fermions at (inverse) temperature $\beta$, we first compute the partition function $Z_N$ of the system in the canonical ensemble via the recursion

formula [39]

$$Z_N = \frac{1}{N} \sum_{\ell=1}^{N} (-1)^{\ell-1} z_\ell Z_{N-\ell}, \tag{B.1}$$

where $Z_0 = 1$ and $z_\ell = \sum_{\boldsymbol{k}} \exp\left(-\ell\beta \frac{\hbar^2 \boldsymbol{k}^2}{2m}\right)$ is the single-particle partition function at temperature $\ell\beta$. Note that Eq. (B.1) involves adding exponentially small numbers with alternating signs, thus one needs to use high-precision arithmetics to obtain reliable results for large particle number $N$.

After obtaining $Z_N$, we evaluate the entropy per particle of ideal Fermi gas using the standard formula

$$s_0 = \frac{1}{N} \left( \ln Z_N - \beta \frac{\partial}{\partial \beta} \ln Z_N \right). \tag{B.2}$$

The derivative in Eq. (B.2) can be conveniently computed by automatic differentiation through high-precision arithmetics, which is natively supported in, e.g., `Julia`[74]. Alternatively, one can manually differentiate both sides of Eq. (B.1) with respect to $\beta$ to derive a similar recursion relation for the energy $E_N = -\frac{\partial}{\partial \beta} \ln Z_N$:

$$E_N = \frac{1}{Z_N} \frac{1}{N} \sum_{\ell=1}^{N} (-1)^{\ell-1} z_\ell Z_{N-\ell} (\ell e_\ell + E_{N-\ell}), \tag{B.3}$$

where $e_\ell = -\frac{\partial}{\partial(\ell\beta)} \ln z_\ell$ is the expected single-particle energy at temperature $\ell\beta$. The starting point of this recursion is, unsurprisingly, $E_0 = 0$.

## C   Twist-averaged boundary conditions

In this work, we aim to simulate an interacting Fermi liquid consisting of finite number of electrons in a finite box. Under the conventional periodic boundary conditions (PBC), the single-particle momenta reside on a discrete lattice $\boldsymbol{k} = \frac{2\pi \boldsymbol{n}}{L}$ ($\boldsymbol{n} \in \mathbb{Z}^2$), as illustrated in Fig. 1(a) of the main text, and the identification of a sharp spherical Fermi surface characteristic of the system is ambiguous. This is a major contribution to the finite size errors of various physical quantities.

A useful technique to alleviate the finite size effect is to use the twist-averaged boundary conditions (TABC) [48]. This amounts to averaging physical observables over a twist vector $\boldsymbol{\theta}_t \in [-\pi, \pi]^2$, which corresponds to the extra phase picked up when the electrons wrap around the periodic boundaries of simulation box. Consequently, the single-particle momenta $\boldsymbol{k} = \frac{1}{L}(2\pi \boldsymbol{n} + \boldsymbol{\theta}_t)$ would be shifted away from the integer lattice points. When the twist average is performed, the effective state occupation "smears out" continuously as in the thermodynamic limit, thus yields better scaling behavior for various physical quantities.

To illustrate the impact of TABC on the simulation, we compute the entropies per particle $s_0$ of two-dimensional ideal Fermi gas at the temperature $T/T_F = 0.15$ for various particle numbers $N$, as shown in Fig. S2. The results labeled as "PBC" are computed at the $\Gamma$ point $\boldsymbol{\theta}_t = \boldsymbol{0}$, whereas the "TABC" results are obtained by averaging the entropy over 10000 uniformly sampled twists. It is clear that TABC results in more regular scaling behavior for small $N$ and converges more smoothly to the thermodynamic limit $N = \infty$. We also plot the same data versus inverse particle number $N^{-1}$ in the inset to better visualize how they extrapolate to the thermodynamic limit.

In practice, we choose to implement the twist average over a $2 \times 2$ Monkhorst-Pack grid [75], which corresponds to a single twist vector $\boldsymbol{\theta}_t = (\frac{\pi}{2}, \frac{\pi}{2})$ [76, 77] after taking into account the point group symmetry of the simulation box. Such a scheme is more convenient

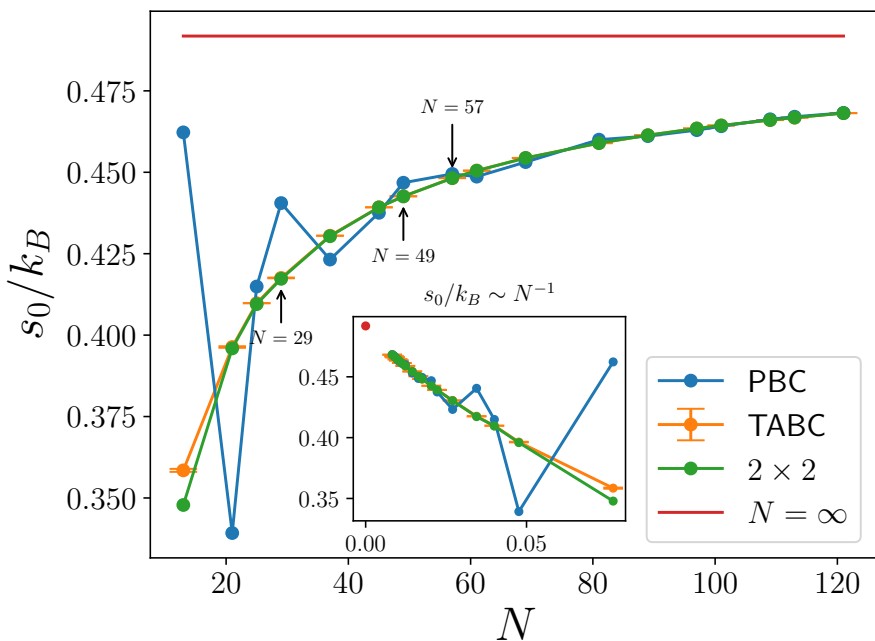

Figure S2: Scaling behavior of the entropy per particle of two-dimensional free Fermi gas at $T/T_F = 0.15$ for various closed-shell particle numbers $N$ from 13 up to 121, as well as the value in the thermodynamic limit $N = \infty$. The points $N = 29, 49$ and 57 used for the calculation of quasiparticle effective mass in this work are specifically annotated. See text for detailed explanations of the legend.

than randomly sampling the twist vectors, and introduces essentially no extra computational cost and code development efforts. We also plot the non-interacting entropies per particle under this scheme in Fig. S2 labeled as "$2 \times 2$", which are in excellent agreement with the results obtained by random sampling. For the largest system size ($N = 57$ electrons) employed in our calculation of the quasiparticle effective mass, the non-interacting entropy deviates from the thermodynamic limit value by about 8%. One would then reasonably expect the uncertainties in the final estimate of effective mass to be within the same level, assuming a similar $N$-scaling behavior of the interacting entropy.

## D Model architectures

This section summarizes the network architectures used for the autoregressive model and normalizing flow. They are adapted from the transformer [36] and FermiNet [47], respectively. Please refer to the original publications for more background on these models.

### D.1 Autoregressive model for $p(K)$

In Algorithm 1, given the momenta $\boldsymbol{k}_1, \boldsymbol{k}_2, \ldots, \boldsymbol{k}_N$ occupied by the $N$ electrons, `CausalTransformer` [36] outputs $N$ $M$-dimensional arrays $\hat{\boldsymbol{k}}_1, \hat{\boldsymbol{k}}_2, \ldots, \hat{\boldsymbol{k}}_N$, where $\hat{\boldsymbol{k}}_i$ denotes the logits of the index of $\boldsymbol{k}_i$ corresponding to the conditional probability $p(\boldsymbol{k}_i | \boldsymbol{k}_{<i})$. The number $M$ of available momenta $\boldsymbol{k} = \frac{1}{L}(2\pi\boldsymbol{n} + \boldsymbol{\theta}_t)$ is determined by a single-particle energy cutoff $|\boldsymbol{n}|^2 \leq$ `Emax`. For $N = 29, 49$ and 57 in our calculations, `Emax` is chosen to be $25, 36$ and 49, respectively, corresponding to the number of available momenta $M = 81, 113$ and 149.

Table S1: The hyperparameters of the causal transformer adopted in this work. Note that we choose the non-linear activation of the fully connected neural network involved in the architecture to be `tanh`, which is smooth enough to avoid potential issues upon automatic differentiation.

| Hyperparameter | Description | Value |
|---|---|---|
| `nlayers` | Number of layers | 2 |
| `modelsize` | The size of input and output, also known as the "embedding dimension" | 16 |
| `nheads` | Number of heads of the self-attention part within each layer | 4 |
| `nhidden` | Number of hidden units of the fully connected neural network within each layer | 32 |

---

**Algorithm 1** Autoregressive probabilistic model for a set of momenta.

---

**Input:** A set of momenta $K = \{k_1, k_2, \ldots, k_N\}$. An index function that identifies each available momentum $k$ to an integer $\mathrm{idx}(k) \in \{1, \ldots, M\}$.

**Output:** Log-likelihood $\ln p(K)$.

1: $\hat{k}_1, \hat{k}_2, \ldots, \hat{k}_N = \mathtt{CausalTransformer}(k_1, k_2, \ldots, k_N)$
2: **for each** $i$ **do**                                                   ▷ ensure Pauli exclusion principle
3:     $\hat{k}_{i, \geq M-N+i+1} = -\infty$
4:     $\hat{k}_{i, \leq \mathrm{idx}(k_{i-1})} = -\infty$
5:     $\hat{k}_i = \hat{k}_i - \mathtt{logsumexp}(\hat{k}_i)$                 ▷ normalization
6: **end for**
7: **return** $\sum_i \hat{k}_{i, \mathrm{idx}(k_i)}$

---

Table S1 summarizes the adopted value of hyperparameters of the network `CausalTransformer` throughout our calculations. For more implementation details, please refer to our source code.[6] Other architectures of the autoregressive model are also possible, such as the masked autoencoder [78], but our choice here based on the transformer turns out to scale more favorably to large systems regarding the number of trainable parameters involved.

## D.2 Normalizing flow for $\Psi_K(R)$

Recall that our goal is not to model a single wavefunction as done in ground-state calculations, but an exponentially large family of orthonormal many-electron basis states $\Psi_K(R)$. As shown in Eq. (8) of the main text, we achieve this goal by bijectively mapping the original electron coordinates $R$ to the quasiparticle coordinates $\zeta$.

---

**Algorithm 2** Translation and permutation equivariant coordinate transformation.

---

**Input:** Electron coordinates $R = \{r_1, r_2, \ldots, r_N\}$ and box length $L$.
**Output:** Transformed quasiparticle coordinates $\zeta$.

1:  $f_1 = \texttt{zeros\_like}(R)$                                                 ▷ array shape: $(N, 2)$
2:  $r_{ij} = r_i - r_j$                                                 ▷ array shape: $(N, N, 2)$
3:  $f_2 = \left[ \sqrt{\sin^2\left(\frac{\pi x_{ij}}{L}\right) + \sin^2\left(\frac{\pi y_{ij}}{L}\right)}, \cos\left(\frac{2\pi r_{ij}}{L}\right), \sin\left(\frac{2\pi r_{ij}}{L}\right) \right]$
4:  **for** $\ell = 1, \cdots, d$ **do**
5:      $g = [f_1, \texttt{mean}(f_1, \texttt{axis} = 0), \texttt{mean}(f_2, \texttt{axis} = 1)]$
6:      **if** $\ell = 1$ **then**
7:          $f_1 = \texttt{softplus}(\texttt{FC}^\ell_{h_1}(g))$                ▷ array shape: $(N, h_1)$
8:          $f_2 = \texttt{softplus}(\texttt{FC}^\ell_{h_2}(f_2))$                ▷ array shape: $(N, N, h_2)$
9:      **else**
10:          $f_1 = \texttt{softplus}(\texttt{FC}^\ell_{h_1}(g)) + f_1$
11:          $f_2 = \texttt{softplus}(\texttt{FC}^\ell_{h_2}(f_2)) + f_2$
12:      **end if**
13: **end for**
14: $g = [f_1, \texttt{mean}(f_1, \texttt{axis} = 0), \texttt{mean}(f_2, \texttt{axis} = 1)]$
15: $f_1 = \texttt{softplus}(\texttt{FC}_{h_1}(g)) + f_1$
16: **return** $R + \texttt{FC}_2(f_1)$

---

Algorithm 2 implements this transformation using blocks from the FermiNet [47]. The one- and two-electron features $f_1$ and $f_2$ are fed into a sequence of fully connected layers $\texttt{FC}^\ell_{h_1}$ and $\texttt{FC}^\ell_{h_2}$, where $h_1, h_2$ are the one- and two-particle feature size, respectively, and $\ell$ ranges from 1 to $d$. Initially, $f_2$ contains pairwise distance features of the electrons in a periodic box [79], while $f_1$ is set to be zero to guarantee the translation equivariance property. Each fully connected layer involved in the algorithm has its own independent parameters, including those at the final stage. Throughout our calculations, the network depth is set to be $d = 2$ and $h_1 = h_2 = 16$. We note that one can repeat Alg. 2 several times to compose an iterative-backflow-like transformation [31, 80] in terms of neural networks.

In principle, one may want to ensure invertibility of the coordinate transformation [66] to make for a genuine normalizing flow model. To our best knowledge, there lacks such a rigorous guarantee for Alg. 2 described above. Nevertheless, the practical training process still appears stable. This is probably because the network we adopt is not very deep.

## E   Details of the training procedure

We sample electron momenta $K$ directly from the autoregressive model in an ancestral manner; see Eq. (6) in the main text. Given the momenta, we then sample the electron coordinates $R$ using the Metropolis algorithm according to Born probability of the wavefunction Eq. (8). The batch size is set to be 8192. We compute the Jacobian $\frac{\partial \zeta}{\partial R}$ of the coordinate transformation involved in the wavefunction ansatz using forward-mode automatic differentiation in Jax [81].

Building on these self-generated samples, we train the autoregressive model Eq. (6) and the normalizing flow Eq. (8) jointly to minimize the variational free energy Eq. (5). The gradient estimators with respect to the parameters $\phi$ and $\theta$ in the autoregressive model and

normalizing flow, respectively, can be easily derived as follows:

$$\nabla_{\phi} F = \mathop{\mathbb{E}}_{K \sim p(K)} \left[ \nabla_{\phi} \ln p(K) \left( \frac{1}{\beta} \ln p(K) + \mathop{\mathbb{E}}_{R \sim |\Psi_K(R)|^2} \left[ E_K^{\text{loc}}(R) \right] \right) \right], \qquad (E.1a)$$

$$\nabla_{\theta} F = 2\Re \mathop{\mathbb{E}}_{K \sim p(K)} \mathop{\mathbb{E}}_{R \sim |\Psi_K(R)|^2} \left[ \nabla_{\theta} \ln \Psi_K^*(R) \cdot E_K^{\text{loc}}(R) \right], \qquad (E.1b)$$

where the wavefunction $\Psi_K(R)$ has been assumed to be complex-valued, as suggested by Eq. (8). One can employ the control variate method [31,34,82] to further reduce the variance of these estimators.

Below we present some more techniques employed to make the training as efficient as possible.

## E.1 The Hutchinson's trick

The computational bottleneck of the variational free energy Eq. (5) and gradient estimators Eq. (E.1) lies in the Laplacian $\nabla^2 \ln \Psi_K(R)$ involved in the local energy, which amounts to computing the trace of the $2N \times 2N$ Hessian matrix $H(\ln \Psi_K(R))$. In the standard automatic differentiation approach, one needs to iterate over the rows or columns of the Hessian, which can be inefficient for large systems.

To reduce the computational complexity, we employ the Hutchinson's stochastic trace estimator [83]

$$\nabla^2 \ln \Psi_K(R) = \mathop{\mathbb{E}}_{\epsilon \sim f(\epsilon)} \left[ \epsilon^T \cdot H(\ln \Psi_K(R)) \cdot \epsilon \right] \qquad (E.2)$$

over a $2N$-dimensional random vector $\epsilon$ with zero mean and identity covariance matrix. The probability density $f(\epsilon)$ can, for example, be chosen as a standard Gaussian. The Hessian-vector product involved in Eq. (E.2) can be efficiently computed by combining forward- and reverse-mode automatic differentiation in Jax [81]. The price we pay, however, is an additional source of randomness on top of the original estimators Eqs. (5) and (E.1), which may potentially require more samples to achieve a given statistical accuracy.

In practice, we choose to apply the Hutchinson's trick only to Hessian of the Jacobian determinant term in $\ln \Psi_K(R)$; see Eq. (8). In this way, one can enjoy an overall speedup of roughly one order of magnitude without sacrificing accuracy due to enlarged variance of the estimators.

## E.2 Stochastic reconfiguration for density matrices

The quantity of central interest for the purpose of this work is the thermal entropy, which turns out to be fairly sensitive to the training process. We thus employ the stochastic reconfiguration method [79], which is much more efficient than conventional first-order optimizers like Adam.

In the context of machine learning a classical generative model or the ground-state variational Monte Carlo of quantum systems, the conventional metric for the parameter space involved is well known as the Fisher information. To find an appropriate metric for the present quantum statistical mechanics setting, the arguably most natural candidate is the Bures distance, defined for two density matrices $\rho$ and $\sigma$ as [84]

$$d_B^2(\rho, \sigma) = 2 \left( 1 - \text{Tr} \sqrt{\sqrt{\rho} \sigma \sqrt{\rho}} \right). \qquad (E.3)$$

The second term on the right-hand side is well known as the quantum fidelity [85].

Recall from Eq. (4) in the main text that $\rho(\phi, \theta) = \sum_K p(K; \phi) |\Psi_K(\theta)\rangle\langle\Psi_K(\theta)|$, where $\phi$ and $\theta$ are the variational parameters in the classical Boltzmann distribution and quantum

many-body basis, respectively. By expanding the Bures distance in the neighborhood of a certain point $(\boldsymbol{\phi}, \boldsymbol{\theta})$, one can obtain [84]

$$d_B^2(\rho(\boldsymbol{\phi}, \boldsymbol{\theta}), \rho(\boldsymbol{\phi} + \Delta\boldsymbol{\phi}, \boldsymbol{\theta} + \Delta\boldsymbol{\theta})) \approx \frac{1}{4}\sum_{ij}\mathcal{I}_{ij}\Delta\phi_i\Delta\phi_j + \sum_{ij}\mathcal{J}_{ij}\Delta\theta_i\Delta\theta_j, \qquad \text{(E.4)}$$

for some positive-definite matrices $\mathcal{I}$ and $\mathcal{J}$. The most important observation is that the desired metric is *block-diagonal* with respect to the two generative models in this work. This fact is favorable in practice, since the metric needs to be inverted to determine the parameter update direction in each training step. Note the size of $\mathcal{I}, \mathcal{J}$ are equal to the number of parameters in the autoregressive model and normalizing flow, respectively, both in the order of several thousand throughout our calculations.

$\mathcal{I}$ coincides exactly with the classical Fisher information matrix

$$\mathcal{I}_{ij} = \sum_K \frac{1}{p(K)}\frac{\partial p(K)}{\partial \phi_i}\frac{\partial p(K)}{\partial \phi_j} = \mathop{\mathbb{E}}_{K \sim p(K)}\left[\frac{\partial \ln p(K)}{\partial \phi_i}\frac{\partial \ln p(K)}{\partial \phi_j}\right], \qquad \text{(E.5)}$$

whereas the quantum component $\mathcal{J}$ reads

$$\mathcal{J}_{ij} = \Re\left(\sum_K p(K)\left\langle\frac{\partial \Psi_K}{\partial \theta_i}\middle|\frac{\partial \Psi_K}{\partial \theta_j}\right\rangle - \sum_{K,K'}\frac{2p(K)p(K')}{p(K)+p(K')}\left\langle\frac{\partial \Psi_K}{\partial \theta_i}\middle|\Psi_{K'}\right\rangle\left\langle\Psi_{K'}\middle|\frac{\partial \Psi_K}{\partial \theta_j}\right\rangle\right). \qquad \text{(E.6)}$$

The double summation over momenta in the second term of Eq. (E.6) is inconvenient to estimate. In practice, we choose to approximate $\mathcal{J}_{ij}$ as the following covariance matrix

$$\begin{aligned}\mathcal{J}_{ij} = \Re\Bigg(&\mathop{\mathbb{E}}_{K \sim p(K)}\mathop{\mathbb{E}}_{R \sim |\Psi_K(R)|^2}\left[\frac{\partial \ln \Psi_K^*(R)}{\partial \theta_i}\frac{\partial \ln \Psi_K(R)}{\partial \theta_j}\right]\\ &- \mathop{\mathbb{E}}_{K \sim p(K)}\mathop{\mathbb{E}}_{R \sim |\Psi_K(R)|^2}\left[\frac{\partial \ln \Psi_K^*(R)}{\partial \theta_i}\right]\mathop{\mathbb{E}}_{K \sim p(K)}\mathop{\mathbb{E}}_{R \sim |\Psi_K(R)|^2}\left[\frac{\partial \ln \Psi_K(R)}{\partial \theta_j}\right]\Bigg).\end{aligned} \qquad \text{(E.7)}$$

Notice the first term is the same as that of Eq. (E.6), which is clearly a natural generalization of the usual quantum Fisher information matrix for pure states. The trainings turn out to still behave quite well.

The update rules for the parameters $\boldsymbol{\phi}, \boldsymbol{\theta}$ read as follows:

$$\Delta\boldsymbol{\phi} = -(\mathcal{I} + \eta\mathbb{1})^{-1}\nabla_{\boldsymbol{\phi}}F, \qquad \text{(E.8a)}$$
$$\Delta\boldsymbol{\theta} = -(\mathcal{J} + \eta\mathbb{1})^{-1}\nabla_{\boldsymbol{\theta}}F, \qquad \text{(E.8b)}$$

where we have added a small shift $\eta = 10^{-3}$ to the diagonal of (modified) Fisher information matrices for numerical stability. The norms of updates are constrained within a threshold of $10^{-3}$ [47], which plays a similar role as the learning rate in other conventional optimizers. See the source code[6] for more details.

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
