# Peer review of "m* of two-dimensional electron gas: A neural canonical transformation study"

_SciPost Physics, doi:SciPost Phys. 14, 154 (2023)_

## Round 1 · Referee Report · Anonymous · 2022-12-21

Strengths
- open source code
- very clear introduction and formulation of the problem
- addresses a classic problem that is very hard to solve yet of high importance
- several original ideas in machine learning; supported by appendices
- authors are very knowledgeable in the methodology
- finite size effects seem to be under good control
Weaknesses
- doubts on the magnitude of the error bars and the control over the error bars
Report
The authors investigate a classic problem: the jellium model, for which they compute the effective mass. Quantum Monte Carlo simulations give conflicting results for this quantity far outside error bars; often the same groups produced inconsistent values over the years. Some of these inconsistencies originate from the different ways in which m* is computed, which rely on certain assumptions that are not always fulfilled. The problem is therefore notoriously hard, and the goal of the present study quite ambitious.
The authors address it by applying the neural canonical transformation approach, which they recently introduced. The density matrix in momentum space is modelled by two generative neutral nets: an autoregressive model for the occupation numbers, and a normalizing flow for the electron coordinates. The authors find values of m* that are substantially lower than what was previously reported in the literature. This is certainly an interesting claim.
My main concern is the error analysis. I find it counterintuitive that the error bars for rs=10 are the smallest. There are no indications in the data leading up to Fig 4 that indicate so, and one may hence worry that the systematic errors are severely underestimated in this work. I therefore ask the authors to carefully check and elaborate their error analysis.
Requested changes
1. The relation m*/m = s /s_0 is based on the validity of Fermi liquid theory. Whereas it is very likely that FL applies to all parameters shown, I wonder if the machine learning data show any deviations from FL theory (ie, to make sure that the data is internally consistent). For instance, do the energies behave as E ~ T^2 for low enough temperature T? Or can we see a plot S(T) for the interacting model?
2. In many figures (Fig 2, 3b, S1) the marker of the data points is big. Could the authors please write the value of the converged answer, with error bars, in the figure caption?
3. The non-monotonicity of the data as a function of N seen in Fig 4 for rs = 3 and 5 seems remarkable to me. What is the explanation? Or is it a consequence of effects seen in FigS2 (which would imply large systematic error bars)?
4.The data shown in Fig 3 seems to fluctuate a lot. Can the authors indicate how they extract the final entropies and error bars from these curves. Naively, the data fluctuate more than 15% with strong autocorrelations extending over many epochs, perhaps even drifting, and this is hard to reconcile with the rather tiny error bars in Fig 4. The authors should provide a more detailed error analysis than the few sentences that are currently written in the text.
5. The energy shown in Fig 2b for rs = 5 goes below the value of the energy reported in the literature whereas the energy for rs = 10 in Fig S1 seems to agree. Could the authors elaborate more? Is there a systematic trend where the method introduced here performs better than other methods? Knowing (ground) state energies as a function of rs would certainly also be a plot of interest.
6. What possibilities exist to compute other, common Fermi liquid parameters?
7. I see no particular reason why a short-range potential (like a Yukawa potential) cannot be studied in the current approach. Is there a particular reason why the authors stayed away from such simpler problems?
Please refer to the attached file reply.pdf for responses to both you and the other referee.
We have also included a list of changes to the manuscript for your convenience.
Thank you!
Author: Hao Xie on 2023-02-10 [id 3338]
(in reply to Report 2 on 2023-01-19)Please refer to the attached file reply.pdf for responses to both you and the other referee.
We have also included a list of changes to the manuscript for your convenience.
Thank you!
Attachment:
reply.pdf

---

## Round 1 · Referee Report · Anonymous · 2023-1-19

Report
In this very interesting work, the authors apply their original, recently introduced neural canonical transformation approach to the problem of the uniform electron gas in two dimensions. Specifically, they calculate the quasiparticle effective mass, finding it substantially lower compared to all the results available in the literature. This finding is especially important given that the available results for the effective mass are rather controversial.
I find the new method very promising and will be ready to recommend the paper for publishing after the authors address my remarks concerning their data.
Remarks
The authors successfully benchmark their method against the ideal gas. Equally (if not more) important would be a benchmark against asymptotically exact analytic results for the (weakly interacting) small-r_s regime, the green dashed line in Fig. 4. While the data for the largest studied system size N=57 appears to be perfectly consistent with the small-r_s analytic curve, this might be a mere illusion given the substantial drift of the data with N. Especially worrisome is the fact that the character of this drift dramatically changes with r_s (cf. corresponding remark by Referee 1).
Based on the above-mentioned circumstances, I suggest that the authors to produce more data:
1. For r_s = 0.5 and 0.25.
2. For N substantially larger than N=57 (if possible; if not, then explain why).

---

## Round 2 · Referee Report · Anonymous (Referee 1) · 2023-3-8

Strengths

(see my previous report)

Weaknesses

(see my previous report)

Report

I am in general satisfied with the revisions and the replies made by the authors, as well as the updates to the open source repo.
As I wrote before, it is a challenging problem and the application of ML to it is certainly in the high risk category, but I fully endorse such non-trivial studies.
I am nevertheless still surprised by the non-monotonicity of the data with N in the low rs regime. The authors mention that a finite size analysis of the interacting model does not exist, but I recall from various Monte Carlo approaches that the extrapolation to the thermodynamic limit is usually under a (surprisingly) good control despite the low particle numbers involved. I will give the authors the benefit of the doubt.

---

## Round 2 · Referee Report · Anonymous (Referee 2) · 2023-3-18

Report

I am satisfied with authors' response to the critical remarks and with the revisions made.

I recommend publishing the manuscript .

---

## Round 2 · List of Changes

1. Update Fig. 4 to include data for rs = 0.5 and 0.25.
  2. Report relevant benchmark values in the caption of Fig. 2 and S1.
  3. Clarify the error analysis of effective mass from the original data; add data processing scripts to the public code repository for reproduction of the final results.
  4. Make slight modifications to some phrases and sentences.

---

## Editorial Decision

published